# The Role of Prophage ϕSa3 in the Adaption of *Staphylococcus aureus* ST398 Sublineages from Human to Animal Hosts [note 1]

**DOI:** 10.3390/antibiotics13020112

**Published:** 2024-01-23

**Authors:** Habib Dastmalchi Saei, Jo-Ann McClure, Ayesha Kashif, Sidong Chen, John M. Conly, Kunyan Zhang

**Affiliations:** 1Department of Pathology & Laboratory Medicine, University of Calgary and Alberta Health Services, Calgary, AB T2N 4N1, Canada; 2Department of Microbiology, Faculty of Veterinary Medicine, Urmia University, Urmia 5756151818, Iran; 3Centre for Antimicrobial Resistance, Alberta Health Services/Alberta Precision Laboratories/University of Calgary, Calgary, AB T2N 4N1, Canada; 4Department of Epidemiology, Public Health College, Guangdong Pharmaceutical University, Guangzhou 510006, China; 5Department of Microbiology, Immunology and Infectious Diseases, University of Calgary, Calgary, AB T2N 4N1, Canada; 6Department of Medicine, University of Calgary and Alberta Health Services, Calgary, AB T2N 4N1, Canada; 7The Calvin, Phoebe and Joan Snyder Institute for Chronic Diseases, University of Calgary and Alberta Health Services, Calgary, AB T2N 4N1, Canada

**Keywords:** *Staphylococcus aureus*, multilocus sequence type (ST), livestock-associated *Staphylococcus aureus* ST398, strain lineage, whole genome sequences (WGS), prophage ϕSa3

## Abstract

*Staphylococcus aureus* sequence type (ST) 398 is a lineage affecting both humans and livestock worldwide. However, the mechanisms underlying its clonal evolution are still not clearly elucidated. We applied whole-genome sequencing (WGS) typing to 45 *S. aureus* strains from China and Canada between 2005 and 2014, in order to gain insight into their evolutionary pathway. Based on WGS phylogenetic analysis, 42 isolates were assigned to the human-associated clade (I/II-GOI) and 3 isolates to livestock-associated clade (IIa). Phylogeny of ϕSa3 sequences revealed five phage groups (Groups 1–5), with Group 1 carrying ϕSa3-Group 1 (ϕSa3-G1), Group 2 carrying ϕSa3-G2, Group 3 carrying ϕSa3-G3, Group 4 carrying ϕSa3-G4 and Group 5 lacking ϕSa3. ϕSa3-G1 was only found in strains that accounted for the most ancestral human clade I, while ϕSa3-G2, ϕSa3-G3 and ϕSa3-G4 were found restricted to sublineages within clade II-GOI. Some isolates of clade II-GOI were also found to be ϕSa3-negative or resistant to methicillin which are unusual characteristics for human-adapted isolates. This study demonstrated a strong association between phylogenetic grouping and phage type, suggesting an important role of ϕSa3 prophage in the evolution of human-adapted ST398 subclones. In addition, our results suggest that this subclone slowly began to adapt to animal hosts by losing ϕSa3 and acquiring methicillin resistance, which was observed in some strains of human-associated clade II-GOI, an intermediate human to livestock transmission clade.

## 1. Introduction

*Staphylococcus aureus* sequence type 398 (ST398) was initially described in the early 2000s, associated with colonization and/or infection of economically important livestock, with the majority of strains being methicillin resistant (MRSA) and multidrug-resistant (MDR) [1,2,3,4,5,6,7,8]. Human infections with LA-MRSA ST398 were subsequently reported in people working in close contact with livestock [2,7,9,10,11,12,13]; however, an increasing number of ST398 infections were later reported in people without direct contact with livestock [14,15,16,17,18,19].

While whole genome sequencing (WGS) has been used to better understand the epidemiology and origin of *S. aureus* ST398, less is known about factors contributing to its evolution. In one study, analysis showed that the human-adapted ST398 strain constitutes a readily transmissible and clinically important clone that differs significantly at the genome level from its livestock-associated counterpart, which has a lower transfer rate between humans [2,20]. Another study by Price et al. using detailed phylogenetic analysis of the lineage, determined that LA-ST398 likely evolved from an ancestral human associated methicillin-susceptible (MSSA) ST398 clade [21]. Following the jump to livestock the strain acquired resistance to methicillin and tetracycline, underwent rapid adaptation, lost ϕSa3 and showed a decreased capacity for human transmission and infection. A separate study by He et al. looking at the evolution of highly virulent CA-MRSA isolates identified ST398 CA-MRSA for the first time, which developed from human-adapted predecessors [22]. These isolates evolved by acquisition of SCC*mec*, conferring methicillin resistance, without gaining other virulence markers typically associated with CA-MRSA (such as PVL). Finally, Diene et al. have suggested that carriage of multiple prophages (polylysogeny) by ST398 played a role in expanding its virulence and increasing its adaptation to humans [23]. The detailed role each phage plays in evolution, however, has not been well elucidated.

Of the prophages carried by ST398, ϕSa3, belonging to the *Siphoviridae* family, deserves special attention due to the fact that it carries the immune evasion cluster (IEC) [24,25]. This cluster contains human-specific immune evasion proteins such as staphylococcal complement inhibitor (Scn), chemotaxis inhibitory protein (Chp) and staphylokinase (Sak), which contribute to human niche adaptation and human invasive infection [24,25]. We have previously shown that different ST398 sublineages possess variable virulence capacities, and that these differences could be attributed to variations in ϕSa3 presence and composition [26]. It has also been shown that the jump of CC398 MSSA from humans to livestock was accompanied by loss of ϕSa3 phage [21]. While there is evidence that ϕSa3 could be related to ST398 adaptation and evolution, no reports have looked specifically to see if variations in ϕSa3 structure between the ST398 sublineages plays a role. As such, we have gathered a collection of ST398 isolates from Canada and Guangdong, China, from 2005 to 2014, and combined them with previously reported isolates from diverse locations to investigate the role that the ϕSa3 structure may have played in ST398 evolution. Here, we report on the presence of different ϕSa3 phage groups in human-associated ST398 strains, as well as on the gradual change and eventual loss of ϕSa3 during the adaptation to livestock.

## 2. Results

### 2.1. ST398 Molecular Characterization

A total of 45 human ST398 isolates were collected and analyzed, with a summary of the strains and their molecular characterization shown in Figure 1. More detailed clinical information can be found in Appendix A. Nine of the isolates were obtained from Canada, while the remaining thirty-six were obtained from Guangdong, China. One isolate from China (GD5) and two from Canada (08S-0030 and N09-00266) were methicillin resistant, with the remaining forty-two isolates being methicillin-susceptible. All three MRSA were determined to be SCC*mec* type V. Two distantly related isolates from China, GD1539 and GD1428, were PVL(+), while the remaining isolates were PVL(-). Five different *spa* types (t034, t571, t1451, t1250, t011) were identified among the forty-five ST398 isolates examined. The most commonly seen *spa* type was t034 (*n* = 20), followed by t571 (*n* = 19), which together represented 86.7% of the isolates. *spa* type t1451 was seen in four isolates, while t1250 and t011 were each detected in one isolate. There was no apparent association between PFGE pattern and *spa* type.

The isolates were divided into 37 clones, whereby isolates in a clone shared identical PFGE banding patterns as well as all other molecular traits. In general, the isolates showed a large degree of variation, with the majority of clones represented by a single isolate. The exceptions were clones #1, #2, #12, #20 and #30, each of which contained two isolates, and clone #8, which contained four isolates. 

### 2.2. Phylogenetic Analysis

A SNP whole genome phylogenetic tree comparing our isolates to the ones presented by Price et al. and He et al. [21,22] is shown in Figure 2. The majority of our isolates fell into the human-associated clades I and II-GOI. Four of the Canadian isolates (293G, 215N, 232N and 387N) were located in clade I, while three (N09-00266, 08S-0030 and CH-EHF) were located in the animal-associated clades, IIa and IIa1i. Five of our Chinese isolates clustered in clade I; however, the majority of the Chinese isolates were distributed in clade II-GOI. None of our Chinese isolates, and only one of the previously described Chinese isolates (LA-MSSA-5), were found in the animal-associated clade IIa. The clade numbers for each isolate that we collected are summarized in Figure 1. 

All of the isolates in clade I were methicillin-susceptible and, while three of the previously reported isolates were PVL(+), all of our isolates were PVL(-). Collection dates for isolates in the group ranged from 1999 to 2015, with ours being collected from 2010 to 2014. Isolates in the group are predominantly human-associated, with only two previously reported ones being of bovine origin. The second clade into which our isolates fell was II-GOI. Isolates in this clade were predominantly methicillin-susceptible; however, some of the previously reported isolates were methicillin resistant, as was one of our isolates from China (GD5). PVL(+) and PVL(-) isolates were present in this clade, including the only two PVL(+) from our study. Collection dates for isolates in the group ranged from 2002 to 2015, with ours being collected in 2010. Once again, the isolates are predominantly of human origin, with some bovine and a pig exception. The third clade into which our isolates fell is clade IIa. One strain in the clade was methicillin-susceptible, but the remaining ones, including the Canadian isolate (N09-00266), were methicillin resistant. All isolates in the group were PVL(-). The Canadian isolate and the MSSA were human-associated; however, the remaining isolates were animal-associated. Collection dates for isolates in the group ranged from 2003 to 2011 (ours was 2011). Finally, our isolates also clustered in clade IIa1i, which contained a mixture of MRSA and MSSA, with one of our isolates being MRSA (08S-0030) and one MSSA (CF-EHF), and all PVL(-). The majority of isolates in this clade were animal-associated, although there were several that were human-associated, including our two. Collection dates ranged from 2000 to 2009, with ours being 2005 and 2008. 

### 2.3. Prophages in the ST398 Isolates

A more detailed analysis of the ST398 isolates was carried out using in silico phage analysis, with the results summarized in Table 1 and Figure 2. All nine of our isolates in clade I carried ϕSa3, while one isolate (GD487) carried ϕSa5 and four isolates (GD1259, GD1108, 215N and 232N) carried ϕSa9. In clade II-GOI, 23 of our 27 (85.2%) isolates carried ϕSa3; however, four isolates (GD1696, GD1884, GD1677 and GD1025) were negative for the phage. Three isolates in clade II-GOI (GD1428, GD1539 and GD705) also carried ϕSa2, three (GD2000, GD1539, GD1025) carried ϕSa1, one (GD1025) carried ϕSa4, six (GD1025, GD1677, GD1884, GD1696, GD1930, GD1706) carried ϕSa6, two (GD1930, GD1696) carried ϕSa7, one (GD1930) carried ϕSa8, three (GD1706, GD104, GD930) carried ϕSa9, two (GD1696, GD1706) carried Spβ and four (GD2000, GD223, GD1930, GD1025) carried PT1028. Similarly, these phages were sporadically carried by the previously reported strains. The three isolates that belonged to the animal-associated clade IIa (N09-00266 from IIa, along with 08S-0030 and CF-EHF from IIa1i) were all ϕSa3 negative, similar to previously reported strains in the clade. One isolate (N09-00266) carried ϕSa2, while two isolates (N09-00266, CH-EHF) carried ϕSa6, one isolate (N09-00266) carried ϕSa9 and two (08S-0030, CF-EHF) carried PT1028. ϕSa6 and PT1028 were commonly carried by previously described isolates in this clade, with ϕSa9 and Spβ carried by a few. Overall, carriage of phages other than ϕSa3 did not correlate with clades or isolate subsets. 

### 2.4. ϕSa3 and Its Structure Is Associated with ST398 Phylogenetic Groupings

It was noted that most of the isolates in the human-associated clades (I and II-GOI) carried phage ϕSa3, while all but one of the isolates in the animal-associated clades did not; therefore, a more detailed examination of the phage was undertaken. The complete ϕSa3 sequence was extracted from our isolates, as well as from the previously reported isolates whenever possible (when available in the GenBank databases). SNP phylogenetic analysis of the phages revealed that they subdivided into five groups: Group 1 carrying ϕSa3-G1, Group 2 carrying ϕSa3-G2, Group 3 carrying ϕSa3-G3, Group 4 carrying ϕSa3-G4 (as shown in Figure 3A) and Group 5 for strains lacking the phage. All the Canadian isolates that carried ϕSa3 (215N, 232N, 293G, 387N), as along with five of the Chinese isolates (GD487, GD1095, GD1108, GD1211, GD1259), carried ϕSa3-G1. Seven isolates from China, including GD399, GD1517, GD1449, GD149, GD1703, GD1706 and GD705, contained ϕSa3-G2. Seven of the Chinese isolates (GD1067, GD1088, GD104, GD1616, GD33, GD930, GD1853) carried ϕSa3-G3 and an additional seven Chinese isolates (GD1428, GD2000, GD1539, GD1042, GD53.1, GD5, GD223) carried ϕSa3-G4. Finally, three Canadian isolates (N09-00266, 08S-0030, CF-EHF) and four Chinese isolates (GD1696, GD1884, GD1677, GD1025) lacked ϕSa3, therefore belonged to phage Group 5.

A closer look was taken of the phage structure for each isolates, with virulence gene content and phage attachment site sequence is presented in Table 2 and Figure 3B. Group 1 phages were quite similar in structure, with the exception of GD487. The majority of phages in this group carried the *scn* and *chp* genes, and shared identical phage attachment sequences. GD487, the one exception, had a significantly different phage attachment sequence, as well as carried the *scn*, *sak* and *sea* genes. Group 2 phages had the most varied structures, and attachment site sequences. The majority of phages in the group carried only the *scn* gene; however, GD1706 carried *scn* and *chp*, while GD705 carried *scn*, *chp*, *sak* and *sea*. Group 3 phages were the most uniform in their structure, all carrying the *scn* and *sak* genes, and all having identical attachment sequences. GD1067 and GD1088 were the most different phages in Group 3, carrying the *chp* gene on top of *scn* and *sak*. Group 4 phages showed more variability in structure and attachment site sequence; however, all the phages carried the *scn*, *chp* and *sak* genes.

To determine if there was any association between the phage structure and phylogenetic clades, the phage group assignation and virulence gene content was included in the whole genome phylogenetic tree in Figure 2. Isolates in clade I carried ϕSa3-G1, most with *scn* and *chp*. Isolates in the lowest branch of clade I, including GD487, differed more in their structure and virulence gene carriage, which is also evident from Figure 3A. Clade II-GOI could be divided into two main branches; isolates in the lower branch, including GD1616, GD33, GD930, GD104, GD1853, GD1088 and GD1067, carried ϕSa3-G3 with *scn* and *sak*. Isolates GD1067 and GD1088 were a bit more distantly related in the branch, and contained ϕSa3-G3 with *scn*, *chp* and *sak*. Isolates in the upper branch of clade II-GOI carried both ϕSa3-G2 and ϕSa3-G4; however, the topmost part of the clade all carried ϕSa3-G4 with *scn*, *chp* and *sak*. As mentioned, none of the isolates in clade IIa carried ϕSa3. 

As there seemed to be an association between the phage type and strain phylogenetic group, we constructed a second WGS-based phylogenetic tree using the same isolates that were used to create the ϕSa3 sequence-based phylogenetic tree (Figure 3C). Comparing the phage ϕSa3 phylogenetic tree (Figure 3A) to the whole genome phylogenetic tree (Figure 3C) showed that there was in fact a strong association between them. Isolates clustered into the same four groups as seen in the phage tree, although the order of isolates within the groups may have differed. 

### 2.5. Methicillin and Tetracycline Resistance in ST398

An examination of the whole genome SNP phylogenetic tree revealed that human-associated ST398 clades were predominantly methicillin-susceptible, while the animal-associated clades were predominantly methicillin resistant. All Isolates in clade I were methicillin susceptible, as were all but nine isolates in clade II-GOI. In clade II-GOI, only one of the nine resistant isolates (GD5) was from our study, and six of the resistant isolates were in the upper part of the clade. While clade IIa-GOI was all methicillin-susceptible animal-associated isolates, the majority of the animal associated isolates in clades, IIa2, IIa1ii, IIa1 and IIa1i were methicillin resistant. This included our isolates N09-00266 in clade IIa and 08S-0030 in clade IIa1i. Similar to methicillin resistance, resistance to tetracycline (*tetM*) was associated with the animal-associated clades, while being completely absent from the human-associated clades. All three of our isolates in the animal-associated clades were *tetM* positive, with N09-00266 also carrying *tetK*. Only one previously reported isolate in the clade was tetracycline sensitive. Interestingly, 12 livestock associated MSSA were previously described that clustered in clade II-GOI and two that clustered in clade I, but all were tetracycline sensitive.

## 3. Discussion

The ancestry and evolution of *Staphylococcus aureus* ST398 has garnered attention as the strain represents an important livestock-associated colonizer/pathogen. The strain also represents a human-associated pathogen that causes infection even in the absence of livestock exposure, believed to have increased pathogenicity specifically to humans [15,20,27,28]. Price at al. have suggested that ST398 first appeared as a human-associated MSSA basal clade, which then evolved into animal-associated lineages through acquisition of SCC*mec* and tetracycline resistance and, notably, loss of ϕSa3 [21]. 

By examining the genetic background and ϕSa3 structure from a collection of ST398 strains isolated in China and Canada from diverse locations over many years, we were able to provide the detailed genetic events that may have driven this evolution from human to animal sublineages. This is the first study that looks at the link between ϕSa3 and evolution of the ST398 clade lineage.

Initial phylogenetic analysis of our isolates (in combination with previously published ST398 isolates) was in agreement with Price et al.’s assertion that ST398 strains fall into distinct human- and livestock-adapted clades, with the human-associated clades appearing earlier in the tree, and the animal-associated clades appearing later [21]. Likewise, in agreement with their study methicillin resistance was gradually gained during the transition from human to animal hosts, as seen in some of the intermediate group II-GOI isolates. Looking more closely at ϕSa3, our results indicated that ϕSa3 structure and content changed over time and could be divided into distinct phage groups, which correlated strongly with ST398 strain whole genome phylogenetic groupings. The existence of different ϕSa3 types has already been reported for ST398 strains, with up to 12 types ranging in size from 40–44 Kb described [29]; however, the strong link between specific phage groups and phylogenetic groups is a new revelation. We saw that as the lineage evolved from the earliest human-associated clade, to the later animal-associated clades, there was a change from ϕSa3-G1 (carrying *scn* and *chp*) in the earliest clade I, to ϕSa3-G3 (carrying *scn* and *sak*), ϕSa3-G2 (carrying *scn*) and ϕSa3-G4 (carrying *scn*, *chp* and *sak*) in the later human-associated clade II-GOI, to phage loss (Group 5) in the animal-associated clades IIa. There was no mixing of phage types within the clade branches, apart from one branch within II-GOI which contained isolates carrying ϕSa3-G2 along with isolates that had lost the phage (Group 5). This observation suggests that it is not only the presence and absence of ϕSa3, but also its structure and gene content playing a part in host adaptation and evolution of human-associated ST398 strains, with the specific complement of human innate immunomodulatory genes carried by each clade likely determined by local selection pressures.

Selection pressures would also account for loss of the human-niche-specific genes in livestock-associated isolates following the adaptation into animals, which has previously been described in ST5 with loss of ϕSa3 following its adaptation to chickens [30]. All of the proteins in this cluster have been shown to have specificity to humans, with Chp showing 30-fold greater activity towards human cells, and Scn unable to inhibit C5b-9 mediated lysis in the presence of sera from other species [25,31]. The fitness costs of maintaining genes/elements which lack functionality in the animal host explains their loss during the transition. Interestingly, four of our ST398-MSSA isolates from the human-associated clade II-GOI were negative for ϕSa3, which is atypical for human-associated isolates. These isolates were seen to cluster closely with previously published ST398 CA-MSSA (human origin) and LA-MSSA (bovine origin) from China [22], which shared similar molecular characteristics. It therefore appears that isolates with this atypical genetic characteristic are circulating widely in the human and animal population of China, with colonization of human hosts by this sublineage, possibly due to the introduction of human-adapted ST398-MSSA isolates (ϕSa3-positive and *tetM*-negative) into livestock, where it subsequently lost the phage and transmitted back to humans, either without acquisition of *tetM*, or with its loss after retransmission to humans.

ϕSa3 may also be influencing ST398 strain evolution by mediating virulence potential of the lineage. In a previous study, we demonstrated that different ST398 sublineages possess variable virulence capacities in a *Caenorhabditis elegans* infection model and that their virulence potential correlated well with the presence or absence, as well as the structure of prophage ϕSa3 [26]. Isolates that were determined to be moderately virulent in the previous study all fell into the earliest human-associated clade and carried ϕSa3-G2 (renamed ϕSa3-G1 in this study due to its more ancestral appearance), while isolates determined to be highly virulent fell into the human-associated clade II-GOI and carried ϕSa3-G1 (renamed ϕSa3-G2 in this study due to its later appearance). Also found in Clade II-GOI were previously reported isolates classified as highly virulent human-adapted CA-MRSA [22], as well as the only methicillin-resistant human-associated isolate from our study (GD5), all of which carried ϕSa3-G4. In their study, He et al. attributed the increase in virulence potential for this subset to acquisition of low-level methicillin resistance SCC*mec* elements [22]. While ST398-MRSA have been reported in cases of severe and sometimes fatal infections in China [22], low virulence colonization isolates of human-adapted ST398-MRSA have also been documented among residents and health care workers in nursing homes in Taiwan [32]. As such, it may not be SCC*mec* that is leading to increased virulence in these isolates, and acquisition of SCC*mec* may simply be the result of local selective pressures favoring antibiotic resistance. The clinical significance of these findings as it relates to invasive infections in humans is difficult to determine with certainty. However, we found that most of the isolates in the human-associated clades (I and II-GOI) carried phage ϕSa3, and as mentioned earlier, it carries the immune evasion cluster (IEC) [24,25] which is an important virulence factor in humans aiding in both colonization and invasive infections [20,33]. A recent paper has supported these findings with respect to phage ϕSa3 in human associated ST398 MSSA strains [34]. The presence of variable lineages of *S. aureus*, typified by ST398, that can crossover to multiple hosts suggests that the ST398 genome, regardless of not harbouring SCC*mec*, can readily adapt to a new host through acquisition of new virulence genes or through gene expression alterations. Expansion of this methicillin-susceptible *S. aureus* ST398 human associated clone into human populations is a definite concern for the future and will need to be monitored carefully. 

Similar selective pressures would have favored acquisition of SCC*mec* in animal-associated lineages, which incidentally possess decreased virulence capacities in humans [2,35]. Instead, the augmented virulence noted for these isolates could be from the specific ϕSa3 type that they carried, namely ϕSa3-G4 bearing three IEC gene, *scn*, *chp* and *sak*. This is in contrast to the other human-associated branches, which only carried two of the genes (*scn*/*chp* or *scn*/*sak*) and had lower virulence. Staphylococcal complement inhibitor (Scn) inhibits central complement convertase and reduces phagocytosis of the opsonized organism, while chemotaxis-inhibiting protein (Chp) reduces leukocyte recruitment and bacterial killing, and Staphylokinase is known to protect *S. aureus* from the human innate immune system and enable penetration of the bacteria into the surrounding tissue [31,36,37]. The presence of all three genes would be expected to heighten virulence in those isolates; however, ϕSa3-4 was not tested in our previous virulence study, and further experiments will be needed to confirm this notion. 

Interestingly, in the current study, we identified three isolates that were collected from humans but belonged to animal-associated clades, each resistant to tetracycline and missing ϕSa3. Two of the isolates were methicillin-resistant and one was susceptible. Cases of livestock-associated ST398 colonization and infection have previously been reported in Canada [38]; however, one of our isolates (CF-EHF) was obtained from a patient attending a Cystic Fibrosis (CF) Clinic. To our knowledge, this is the first report showing colonization with ST398 LA-MSSA in a CF patient in Canada, although CF patients colonized with a ST398 LA-MRSA have been reported in Belgium [39], Brazil [40] and Poland [41]. We also found it noteworthy that, between 2010 and 2014, we isolated nine human-associated ST398-MSSA, each belonging to the most ancestral clade I. Similarly, between 2012 and 2015, He et al. identified several isolates of CA-MSSA and LA-MSSA within that clade [22]. These isolates indicate that the ancestral lineage continues to circulate in humans and animals up to present time in China and Canada. If the evolutionary hypothesis presented by Price et al. [21] was correct, we would expect to see the gradual disappearance of older clades as they were replaced with newer ones, yet these isolates remain one of the dominant lineages in China. As such, one can speculate that this phylogenetic tree may not be representing evolution per se, but may simply be groupings reflecting local selective pressures, heavily influenced by the specific ϕSa3 carried.

As the number of strains and locations were limited in our study, more studies are needed to confirm our findings and hypothesis with stains from other parts of the world, particularly in different settings. The limited diversity in clinical source also makes it difficult to determine if there exists a link between isolate source and phage group, therefore, further studies would be needed in this area as well.

In conclusion, molecular and phylogenetic analysis has shown that, not only do *Staphylococcus aureus* ST398 isolates cluster into human-associated and animal-associated clades, but that ϕSa3 structure and content is seen to change with progression through the clades, before being lost after the transition to animals. The strong association between phylogenetic grouping and phage type suggests that ϕSa3 plays a significant role in the evolution of this lineage. 

## 4. Materials and Methods

### 4.1. Bacterial Strains

All *S. aureus* ST398 isolates collected over a 10-year time frame (2005–2014) in our study areas were included in this analysis. All strains were of human origin, with detailed clinical information found in Appendix A. Strains labelled as GDxxxx were obtained from public community school students or hospital patients during a MSSA/MRSA epidemiological prevalence survey from February to April 2010 in Guangzhou, Guangdong, China. Strains 387T and 387N were obtained from patients attending the CUPS (Calgary Urban Project Society) clinic, while strains 215N, 215W, 293G and 232N were obtained from patients at the STI (sexually transmitted infection) clinic, all in Calgary, AB, Canada, in 2014 [42]. Strain CF-EHF was obtained from a patient attending the Cystic Fibrosis Clinic in Calgary, AB, Canada, in 2005. Strains 08S-0030 and N09-00266 were obtained from the National Microbiology Laboratory in Winnipeg, Manitoba, Canada, in 2008 and 2009, respectively.

### 4.2. Strain Molecular Characterization

Staphylococcal isolates were fingerprinted by pulsed field gel electrophoresis (PFGE) using a modified protocol involving digestion with *Cfr*9I [43,44]. DNA fingerprints were analyzed using BioNumerics Ver. 6.6 (Applied Maths, Sint-Martens-Lattem, Belgium) using a tolerance of 1.5%. An in-house polymerase chain reaction (PCR) assay which detects the *mecA* and PVL genes was used to characterize the isolates [45], followed by staphylococcal protein A (*spa*) typing [46], multilocus sequence typing (MLST) [47] and SCC*mec* typing [48,49].

### 4.3. DNA Sequencing and Whole Genome Sequence Analysis

To study our ST398 isolates more closely, the genome of at least one representative strain in each clone was sequenced. Genomic DNA was isolated from the ST398 strains using phenol:chloroform extraction.

All isolates but 6 were sequenced with Illumina MiSeq technology (2 × 250 bp), with the exceptions being omitted because they were identical on PFGE and in molecular characteristics to another isolate already chosen for sequencing. Strains GD5, GD104, GD487, GD705, GD1025, GD1042, GD1108, GD1259, GD1428, GD1539, GD1677, GD1696, GD1706, GD1930, GD2002, 293G and N09-00266 were further sequenced using Pacific Biosciences (PacBio) RSII sequencing technology (McGill University Génome Québec). Strains 08S-0030, GD33, GD53.1, GD930, GD1025, GD1095, GD1449, GD1616, GD1853, GD1930, GD2000, GD2002 and EH-F were sequenced with a MinION using the ligation sequencing kit and standard conditions (Oxford Nanopore Technologies [ONT], Oxford, UK).

Single nucleotide polymorphism (SNP) whole genome sequence (WGS) phylogenetic analysis was performed using CSI Phylogeny v1.4 with default settings, using strain N315 (BA000018) as the reference genome (Center for Genomic Epidemiology, Lyngby, Denmark). Our isolates were compared with the isolates presented by Price et al. and He et al., which were collected between 1999 and 2015 [21,22]. FigTree v1.4.3 (Institute of Evolutionary Biology, University of Edinburgh, Edinburgh, United Kingdom) was used for phylogenetic tree visualization. Prophage identification and annotation was performed using PHASTER software [50,51], and the comparisons performed using Easyfig [52]. Prophage phylogenetic analysis was performed using CSI Phylogeny v1.4 with default settings, using prophage phi 13 from strain NCTC 8325 (NC_004617) as the reference. Virulence genes were identified with Virulence finder 1.5 [53] and oriTfinder v1.0 [54] and antibiotic resistance determinants were identified with ResFinder 3.2 [55] as well as with oriTfinder v1.0.

## Figures and Tables

**Figure 1 antibiotics-13-00112-f001:**
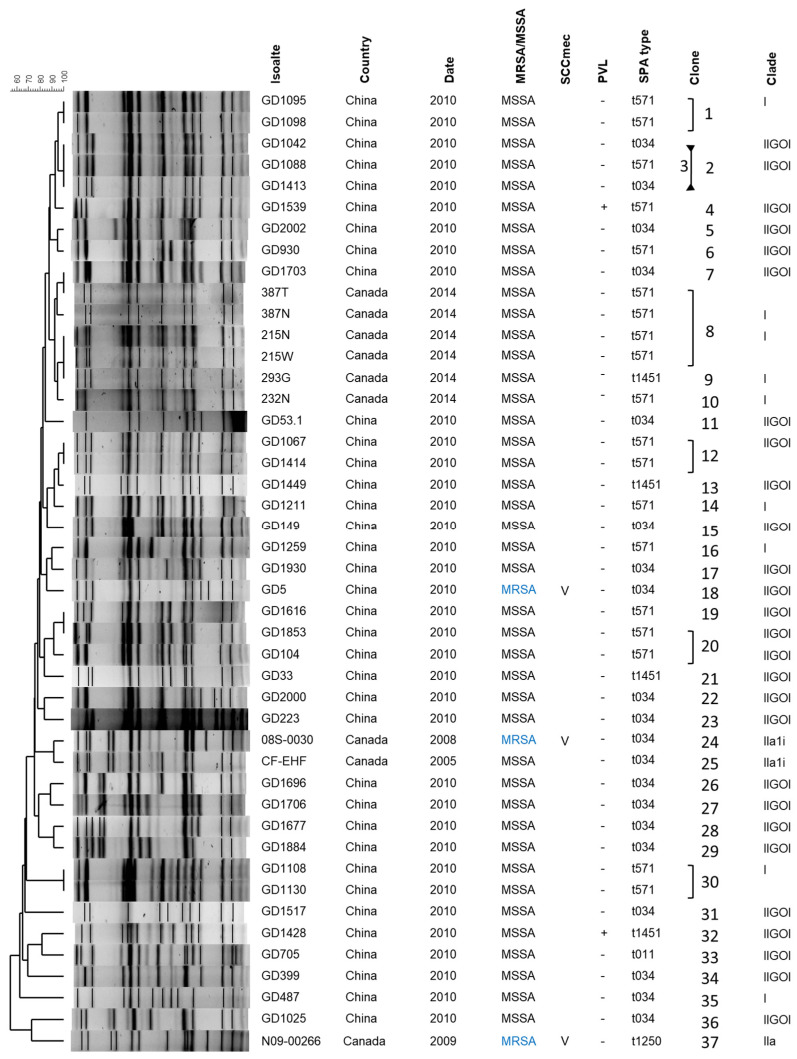
ST398 molecular characterization, with PFGE fingerprint, location and date of isolation, methicillin resistance profile, SCC*mec* type, PVL profile, *spa* type, clonal group and relationship to previously described evolutionary groups [21] included. MSSA, methicillin-susceptible *S. aureus*; MRSA, methicillin-resistant *S. aureus* (blue font); +, gene is present; -, gene is absent.

**Figure 2 antibiotics-13-00112-f002:**
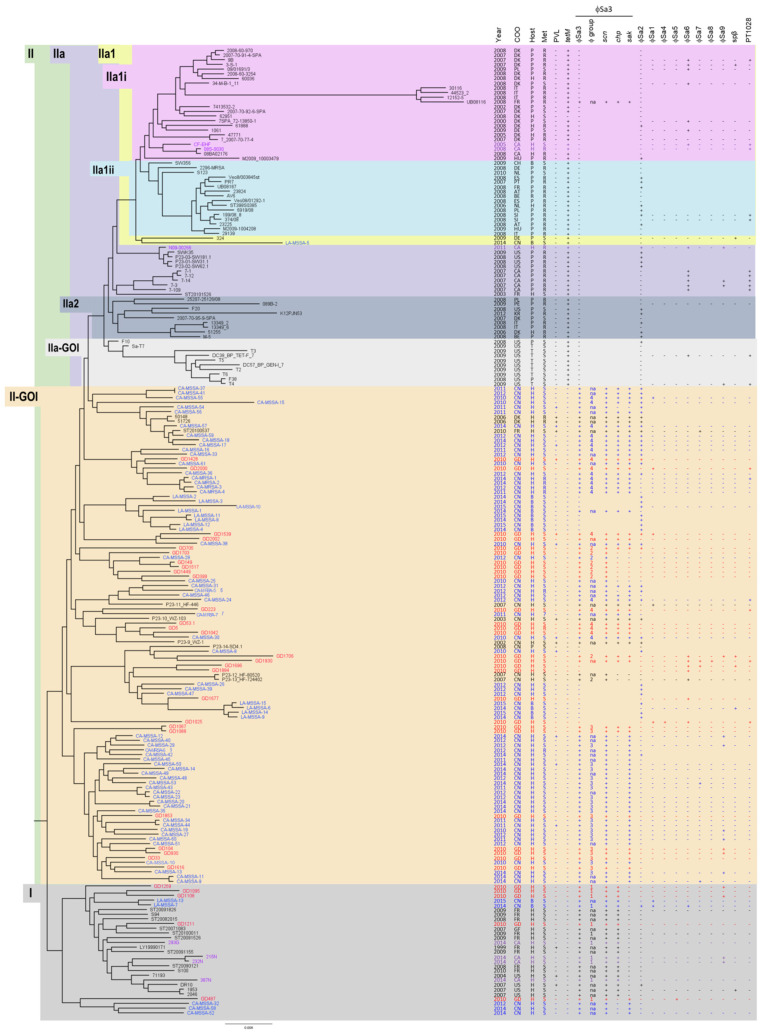
ST398 Phylogenetic tree showing the relationship between our strains and previously described strains, as well as their association with proposed evolutionary clades. Clades and groups were labeled in a hierarchical fashion and the tree was rooted with clade I as described by Price et al. [21]. The Canadian strains are shown in purple font and fell primarily into the oldest clade (I). Chinese strains are shown in red font and fell primarily into the next oldest clade (II-GOI). Canadian strains were also found in the more recent animal associated clades (IIa and IIa1i). Strains from Price et al. are in black font, while strains from He et al. are in blue font [22]. Year of isolation, country of origin (COO), host from which isolate was obtained, methicillin resistance profile (MET), Panton valentine Leukocidin carriage (PVL), tetracycline M resistance (*tet*M), ϕSa3 carriage (including phage group and presence of virulence genes) and other phage carriage are noted on the right. AT, Austria; BE, Belgium; CA, Canada; CH, Switzerland; CN, China; De, Germany; DK, Denmark; ES, Spain; FI, Finland; FR, France; GF, French Guiana; HU, Hungry; IT, Italy; NL, The Netherland; PE, Peru; PL, Poland; PT, Portugal; SI, Slovenian; US, United States; P, Pig; H, Human; R, Horse; T, Turkey; B, Bovine; R, Resistant; S, Susceptible; NA, Not Applicable; *scn*, *Staphylococcus* complement inhibitor; *chp*, chemotaxis inhibitory protein; *sak*, staphylokinase.

**Figure 3 antibiotics-13-00112-f003:**
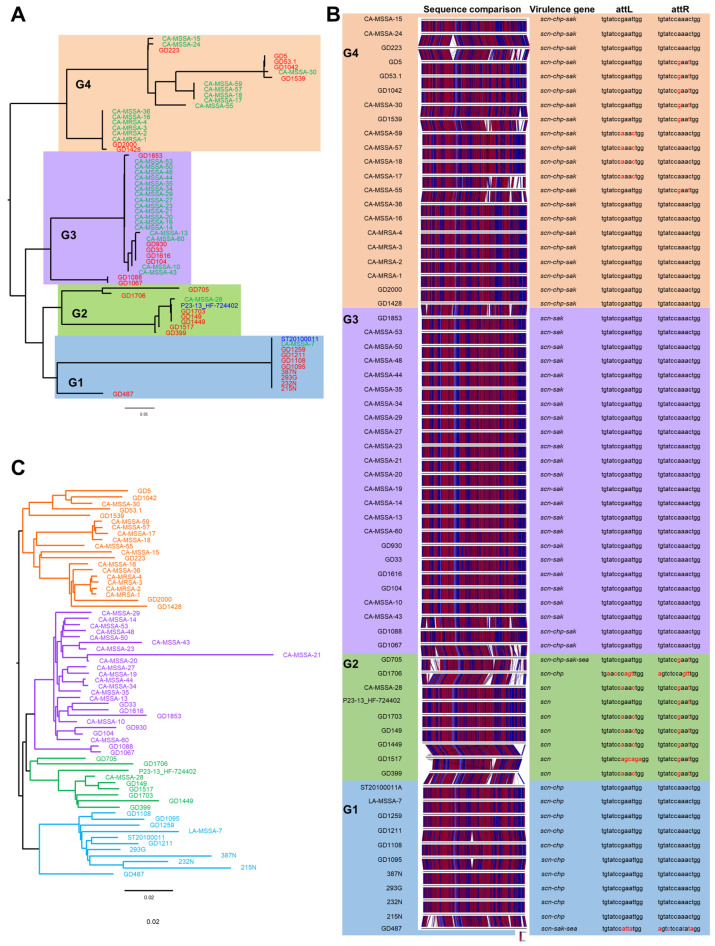
The phylogeny and structure of ϕSa3 in ST398. (**A**) A midpoint rooted SNP phylogenetic dendrogram of the ϕSa3 sequences from attachment site L (attL) to attachment site R (attR) shows that the phages can be divided into 4 groups, ϕSa3-G1, -G2, -G3 and -G4. Blue shading denotes phage group 1 (ϕSa3-G1), green shading denotes phage group 2 (ϕSa3-G2), purple shading denotes phage group 3 (ϕSa3-G3) and orange shading denotes phage group 4 (ϕSa3-G4). Phi 13 (ϕSa3) from strain NCTC8325 (NC_004617) was used as the reference, and the scale bar indicates the number of substitutions per site. Isolates from this study are indicated in red font, strains from the He et al. study [22] in green font, and strains from the Price et al. study [21] in blue font. (**B**) EasyFig comparison of ϕSa3 from the same strains shown in Figure (**A**), in the same order from bottom to top, shows the similarities and differences within and between the phage groups. Phages in group 2 (green shading) showed the most variability, while phages in groups 1 (blue shading) and 3 (purple shading) were much more uniform in structure, and phages in group 4 (orange shading) fell in between. The associated virulence genes, as well as attL and attR sequences are shown. The most common attachment site sequences are shown in black font, with base pairs that differ from the common sequence shown in red. Shading color represents the same phage groups as in (**A**). attL, phage attachment left; attR, phage attachment right; *scn*, *Staphylococcus* complement inhibitor; *chp*, chemotaxis inhibitory protein; *sak*, staphylokinase; *sea*, staphylococcus enterotoxin A. (**C**) SNP whole genome phylogenetic dendrogram of the same strains shown in (**A**) shows that the strains clustered into the identical groups as seen when only ϕSa3 was analysed. The strains are colored to match the shading from the phage groups in (**A**). Strain N315 (BA000018) was used as the reference genome, the genome is midpoint rooted, and the scale bar indicates the number of substitutions per site.

**Table 1 antibiotics-13-00112-t001:** Prophages and genomic elements identified in the ST398 isolates.

Isolate	Clade	ϕSa3	ϕSa1	ϕSa2	ϕSa4	ϕSa5	ϕSa6	ϕSa7	ϕSa8	ϕSa9	ϕSa12	spβ	PT1028
**GD1095**	I	+	-	-	-	-	-	-	-	-	-	-	-
**GD1042**	IIGOI	+	-	-	-	-	-	-	-	-	-	-	-
**GD1088**	IIGOI	+	-	-	-	-	-	-	-	-	-	-	-
**GD1539**	IIGOI	+	+	+	-	-	-	-	-	-	-	-	-
**GD2002**	IIGOI	+	-	-	-	-	-	-	-	-	-	-	-
**GD930**	IIGOI	+	-	-	-	-	-	-	-	+	-	-	-
**GD1703**	IIGOI	+	-	-	-	-	-	-	-	-	-	-	-
**387N**	I	+	-	-	-	-	-	-	-	-	-	-	-
**215N**	I	+	-	-	-	-	-	-	-	+	-	-	-
**293G**	I	+	-	-	-	-	-	-	-	-	-	-	-
**232N**	I	+	-	-	-	-	-	-	-	+	-	-	-
**GD53.1**	IIGOI	+	+	-	-	-	-	-	-	-	-	-	-
**GD1067**	IIGOI	+	-	-	-	-	-	-	-	-	-	-	-
**GD1449**	IIGOI	+	-	-	-	-	-	-	-	-	-	-	-
**GD1211**	I	+	-	-	-	-	-	-	-	-	-	-	-
**GD149**	IIGOI	+	-	-	-	-	-	-	-	-	-	-	-
**GD1259**	I	+	-	-	-	-	-	-	-	+	-	-	-
**GD1930**	IIGOI	+	-	-	-	-	+	+	+	-	-	-	+
**GD5**	IIGOI	+	-	-	-	-	-	-	-	-	-	-	-
**GD1616**	IIGOI	+	-	-	-	-	-	-	-	-	-	-	-
**GD1853**	IIGOI	+	-	-	-	-	-	-	-	-	-	-	-
**GD104**	IIGOI	+	-	-	-	-	-	-	-	+	-	-	-
**GD33**	IIGOI	+	-	-	-	-	-	-	-	-	-	-	-
**GD2000**	IIGOI	+	+	-	-	-	-	-	-	-	-	-	+
**GD223**	IIGOI	+	-	-	-	-	-	-	-	-	-	-	+
**08S-0030**	IIa1i	-	-	-	-	-	-	+	-	-	-	-	+
**CF-EHF**	IIa1i	-	-	-	-	-	+	-	-	-	-	-	+
**GD1696**	IIGOI	-	-	-	-	-	+	+	-	-	-	+	-
**GD1706**	IIGOI	+	-	-	-	-	+	-	-	+	-	+	-
**GD1677**	IIGOI	-	-	-	-	-	+	-	-	-	-	-	-
**GD1884**	IIGOI	-	-	-	-	-	+	-	-	-	-	-	-
**GD1108**	I	+	-	-	-	-	-	-	-	+	-	-	-
**GD1517**	IIGOI	+	-	-	-	-	-	-	-	-	-	-	-
**GD1428**	IIGOI	+	-	+	-	-	-	-	-	-	-	-	-
**GD705**	IIGOI	+	-	+	-	-	-	-	-	-	-	-	-
**GD399**	IIGOI	+	-	-	-	-	-	-	-	-	-	-	-
**GD487**	I	+	-	-	-	+	-	-	-	-	-	-	-
**GD1025**	IIGOI	-	+	-	+	-	+	-	-	-	-	-	+
**N09-00266**	IIa	-	-	+	-	-	+	-	-	+	-	-	-

Note: +, phage present; -, phage not present.

**Table 2 antibiotics-13-00112-t002:** Phage group, virulence gene content and attachment site sequences for the phages present in the ST398.

Isolate	ϕSa3 Group	Virulence Gene				attL	attR
		*scn*	*chp*	*sak*	*sea*		
**GD223**	G4	+	+	+	-	tgtatccgaattgg	tgtatccaaactgg
**GD5**	G4	+	+	+	-	tgtatccgaattgg	tgtatcc**g**aa**t**tgg
**GD53.1**	G4	+	+	+	-	tgtatccgaattgg	tgtatcc**g**aa**t**tgg
**GD1042**	G4	+	+	+	-	tgtatccgaattgg	tgtatcc**g**aa**t**tgg
**GD1539**	G4	+	+	+	-	tgtatccgaattgg	tgtatccgaattgg
**GD2000**	G4	+	+	+	-	tgtatccgaattgg	tgtatccaaactgg
**GD1428**	G4	+	+	+	-	tgtatccgaattgg	tgtatccaaactgg
**GD1853**	G3	+	-	+	-	tgtatccgaattgg	tgtatccaaactgg
**GD930**	G3	+	-	+	-	tgtatccgaattgg	tgtatccaaactgg
**GD33**	G3	+	-	+	-	tgtatccgaattgg	tgtatccaaactgg
**GD1616**	G3	+	-	+	-	tgtatccgaattgg	tgtatccaaactgg
**GD104**	G3	+	-	+	-	tgtatccgaattgg	tgtatccaaactgg
**GD1088**	G3	+	+	+	-	tgtatccgaattgg	tgtatccaaactgg
**GD1067**	G3	+	+	+	-	tgtatccgaattgg	tgtatccaaactgg
**GD705**	G2	+	+	+	+	tgtatccgaattgg	tgtatcc**g**aa**t**tgg
**GD1706**	G2	+	+	-	-	tg**a**a**c**cc**agt**ttgg	**a**gtctcca**gtt**tgg
**GD1703**	G2	+	-	-	-	tgtatcc**a**aa**c**tgg	tgtatcc**g**aa**t**tgg
**GD149**	G2	+	-	-	-	tgtatcc**a**aa**c**tgg	tgtatcc**g**aa**t**tgg
**GD1449**	G2	+	-	-	-	tgtatcc**a**aa**c**tgg	tgtatcc**g**aa**t**tgg
**GD1517**	G2	+	-	-	-	tgtatcc**agcaga**gg	tgtatcc**g**aa**t**tgg
**GD399**	G2	+	-	-	-	tgtatcc**a**aa**c**tgg	tgtatcc**g**aa**t**tgg
**GD1259**	G1	+	+	-	-	tgtatccgaattgg	tgtatccaaactgg
**GD1211**	G1	+	+	-	-	tgtatccgaattgg	tgtatccaaactgg
**GD1108**	G1	+	+	-	-	tgtatccgaattgg	tgtatccaaactgg
**GD1095**	G1	+	+	-	-	tgtatccgaattgg	tgtatccaaactgg
**387N**	G1	+	+	-	-	tgtatccgaattgg	tgtatccaaactgg
**293G**	G1	+	+	-	-	tgtatccgaattgg	tgtatccaaactgg
**232N**	G1	+	+	-	-	tgtatccgaattgg	tgtatccaaactgg
**215N**	G1	+	+	-	-	tgtatccgaattgg	tgtatccaaactgg
**GD487**	G1	+	-	+	+	tgtatcc**atta**tgg	**a**gt**c**tcca**t**a**ta**gg

Note: The most common attachment site sequences are shown in regular font, with base pairs that differ from the common sequence shown in bold font. +, gene present; -, gene absent; attL, phage attachment left; attR, phage attachment right; *scn*, *Staphylococcus* complement inhibitor; *chp*, chemotaxis inhibitory protein; *sak*, staphylokinase; *sea*, staphylococcus enterotoxin A.

## Data Availability

Sequencing reads have been deposited in NCBI under accession numbers CP019591 (293G), CP019593 (GD705), CP019592 (GD5), CP019595 (GD1677), CP019594 (GD1539), CP040232 (GD1706), CP040229 (GD487), CP040230 (GD1108), CP040233 (GD1696), SRX5802346 (GD1517), SRX5802629 (GD399), SRX5802683 (215N), SRX5802701 (232N), SRX5807140 (387N), SRX5807290 (GD1884), and BioProject number PRJNA1055539.

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
