# Peer review of "The Role of Prophage ϕSa3 in the Adaption of Staphylococcus aureus ST398 Sublineages from Human to Animal Hosts†"

_antibiotics, 2024, doi:10.3390/antibiotics13020112_

Round 1

Reviewer 1 Report

Comments and Suggestions for Authors

This manuscript provides a comprehensive analysis of the clonal evolution of Staphylococcus aureus sequence type (ST) 398, shedding light on its transmission between humans and livestock. The study utilizes whole-genome sequencing (WGS) typing on 45 S. aureus strains from China and Canada spanning the years 2005-2014.

The strengths of this manuscript lie in its originality and the valuable insights it offers into the evolutionary dynamics of S. aureus ST398. The differentiation between human-associated and livestock-associated clades, along with the identification of phage groups, is a noteworthy contribution to the field. However, the figures in the abstract lack clarity, impeding a full understanding of the results. Redesigning these figures is crucial for enhancing the overall presentation and comprehension of the data.

Moreover, the discussion could be strengthened by delving deeper into the potential clinical applications of these findings. A more focused exploration of how the discovered results might be applied in a clinical context or guide future research directions would enrich the manuscript.

In conclusion, this manuscript is a valuable addition to the field of Staphylococcus aureus research. The incorporation of clearer figures, along with a more detailed discussion on the clinical implications and future research directions, would significantly enhance its impact. I recommend a minor revision focusing on these aspects to ensure the manuscript's clarity and broader relevance.

Reviewer 2 Report

Comments and Suggestions for Authors

The authors, through their study involving whole genome sequencing of S. aureus ST398 strains and phylogenetic analysis, attempted to demonstrate the correlation of the prophage φSa3 in the evolution of this clone and its potential association with the adaptation of strains to human and animal hosts. Utilizing a comparative analysis with strains from other studies, they managed to observe the possible role of the prophage in the evolution of this clone. As rightly mentioned, there are limitations in the study, such as the number of strains and the restricted collection areas. Nonetheless, it is a well-structured and characterized study with interesting results. A study with more recent strains would be of great interest for a better understanding of the clone's evolution closer to the present day.

During the time frame in which you collected the samples, were there no other ST398 S. aureus strains in the studied areas? If yes, why did you choose only these 42 for your study?

Author Response

Please see the attachment. Thank ;you.

Reviewer 3 Report

Comments and Suggestions for Authors

The manuscript assesses prophage ϕSa3 in the adaption of Staphylococcus aureus ST398 to animal hosts.

Major issues

-Most of the first paragraph of 2.1. is really M&M and thus should be re-arranged appropriately. Moreover, full disclosure of details of the strains must be presented in a summary table in the main text, complemented by all the identification details (origin, source, year of isolation, etc.) in supplementary material.

Noted that avoidance of presenting these details will raise suspicions.

-Please provide the criteria for selection of the strains used in the study.

In all, sub-section 4.1. is not clear and also possibly seems evasive. This is not acceptable.

-Again, the opening passages in 2.2. should be transferred to M&M.

-Was classification based on clinical source of the isolates attempted? I suggest that perhaps that can provide value to the findings.

-Is there a knowledge that people from whom strains were isolated, had an association with animals (pets, livestock)?

Minor issues

-2.3. and 2.4. Please summarise findings in Tables.

-Dividing Discussion in sub-sections will help flow of reading.

Overall. The results are not convincing as they are laid out. I suggest extensive restructuring of the manuscript and resubmission for new evaluation.

Round 2

Reviewer 3 Report

Comments and Suggestions for Authors

The authors have addressed the points raised. No further comments.